# Powering Neural Architecture Search with Robust Masked Autoencoders

## Abstract

Neural Architecture Search (NAS) relies heavily on labeled data, which is labor-intensive and time-consuming to obtain. In this paper, we propose a novel NAS method based on an unsupervised paradigm, specifically Masked Autoencoders (MAE), thereby eliminating the need for labeled data during the searching process. By replacing the supervised learning objective with an image reconstruction task, our approach enables the robust discovery of network architectures without compromising performance and generalization ability. Additionally, we address the problem of performance collapse encountered in the widely-used Differentiable Architecture Search (DARTS) in the unsupervised setting by designing a hierarchical decoder. Through extensive experiments conducted across various search spaces and datasets, we demonstrate the effectiveness and robustness of our method, offering empirical evidence of its superiority over baseline approaches.

## 1 Introduction

In recent years, there has been a significant surge of interest in Neural Architecture Search (NAS) within the machine learning field Zela et al. (2020); Liang et al. (2019); Ramachandran et al. (2018); Liu et al. (2019a). NAS algorithms have emerged as a powerful tool for automatically discovering superior network architectures, potentially saving valuable time and effort for human experts. These algorithms have demonstrated remarkable success in various tasks, including but not limited to image classification and object detection, by discovering neural architectures that achieve state-of-the-art performance.

Existing NAS methods focus on learning from labeled data, leveraging the power of supervised learning to guide the search for the optimal architectures. By utilizing labeled data, consisting of samples paired with their corresponding ground truth labels, NAS algorithms aim to find competitive models that can perform effectively across a variety of tasks and scenarios. However, obtaining substantial quantities of human-annotated data proves to be costly and time-consuming. A portion of the research Liu et al. (2020); Yan et al. (2020); Zhang et al. (2021) has shifted its attention towards exploring methods to minimize the reliance on annotated data.

In this study, we present a novel NAS method called MAE-NAS based on Masked Autoencoders He et al. (2022). To the best of our knowledge, this area has received limited explicit exploration in previous research. Specifically, we apply the unsupervised paradigm of mask image modeling Xie et al. (2022); He et al. (2022) to the widely adopted DARTS method Liu et al. (2019b). Instead of relying on the supervised classification objective employed in DARTS, we replace it with the image reconstruction task, thereby preventing the need for labeled data during the search process. Our approach is shown in Figure 1, where the randomly masked images are fed into the encoder and passed through the decoder to produce a reconstructed image. The encoder section covers the search space for NAS, aimed at selecting a superior model to enhance the quality of the reconstructed image. Based on MAE-NAS, we further investigate the issue of performance collapse of DARTS in an unsupervised setting. The results indicate that the occurrence of collapse is highly correlated with the size of the mask ratio. Notably, a higher mask ratio (i.e., greater than 0.5) effectively enables DARTS to robustly overcome performance collapse. Motivated by this, we propose a hierarchical decoder to stabilize the search process, fundamentally solving the issue of performance collapse in DARTS. Specifically, the decoder takes hierarchical features as input, which encodes both fine and coarse-grained information of the input image, and its output is the reconstructed image.

Figure 1: The framework of MAE-NAS. The input is an image with an applied mask, which is first fed into an encoder and then passed a hierarchical decoder, ultimately producing a reconstructed image. The encoder section covers the search space for NAS, aimed at selecting a superior model to enhance the quality of the reconstructed image.

The effectiveness of our method has been verified on **seven** widely used search spaces and **three** datasets, providing compelling empirical evidence. Experimental results on ImageNet Deng et al. (2009) and MS COCO dataset Lin et al. (2014) demonstrate that MAE-NAS achieves superior performance over its counterparts while adhering to the comparable complexity constraint and the same search space. Furthermore, we have conducted comprehensive experimental analysis and ablation studies on NASBench-201 Dong & Yang (2020), NASBench-101Ying et al. (2019) and TransNas-Bench-101Duan et al. (2021) to gain a deeper understanding of the characteristics of MAE-NAS. These analyses reveal masked autoencoders are robust neural architecture search learners.

In a nutshell, our main contributions can be summarized as follows:

- We present a novel NAS method that leverages masked autoencoders to enable *label-free searching*, which directly addresses the challenge of NAS in scenarios where labeled data is expensive and not readily available.

- The proposed approach is designed to be plug-and-play, seamlessly integrating with the existing supervised NAS methods. In our experiments, we showcase its compatibility with other orthogonal DARTS variants. By removing their handcrafted indicators, MAE-NAS demonstrates its integration ability without incurring additional overhead.

- Our approach achieves better or on-par results with its supervised and unsupervised counterparts, which indicates the applicability of our method in real practice. Importantly, MAE-NAS offers a new perspective on solving the performance collapse issue of DARTS in the unsupervised paradigm.

## 2 RELATED WORK

**Supervised neural architecture search**. It has emerged as a prominent paradigm in NAS research. Initially, NAS methods involved training candidate architectures from scratch and iteratively updating the controller based on performance feedback. However, this approach needs a substantial computational cost, as exemplified by NAS-Net Zoph et al. (2018b), which requires approximately 1350-1800 GPU days. To address this challenge and enhance the efficiency of NAS, weight-sharing mechanisms have been widely adopted in various studies. These approaches can also be classified into two main categories: one-shot methods Bender et al. (2018); Dong & Yang (2019a;a); Li et al. (2020c); Chu et al. (2021b; 2023) and gradient-based methods Liu et al. (2019b); Chu et al. (2021a; 2020); Li et al. (2020a).

One-shot methods Bender et al. (2018); Cai et al. (2019); Chen et al. (2019b) entail training an over-parameterized supernet using diverse sampling strategies. Once the supernet is effectively trained, multiple child models are evaluated as potential alternatives, and those exhibiting superior performance are selected. In contrast, gradient-based algorithms optimize both the network weights and architecture parameters simultaneously through back-propagation. The selection of operators is

based on the magnitudes of the architecture parameters. These approaches aim to reduce the computational cost of NAS while still achieving commendable performance. Through the utilization of weight-sharing mechanisms and the adoption of different optimization strategies, researchers have made significant progress in enhancing the efficiency and practicality of NAS. Given the differentiable and end-to-end characteristics of the DARTS paradigm, our study adopts it to investigate the unsupervised NAS.

**Unsupervised neural architecture search**. Recently, there has been a growing emphasis on the application of unsupervised learning, including the field of NAS. This unsupervised paradigm has gained attention due to its potential to alleviate the reliance on labeled data. Notably, UnNAS Liu et al. (2020) provides a comprehensive analysis of the impact of labeled data on NAS performance. Their findings challenge the conventional belief that labeled data is indispensable for NAS. Building upon this, RLNAS Zhang et al. (2021) leverages random labels instead of true labels. Surprisingly, their research demonstrates that neural architectures discovered using random labels can achieve comparable or even superior performance over supervised NAS methods. MIM-DARTSGuo et al. (2022) introduces the MAE objective as an auxiliary loss on top of the original supervised loss to address the issue of performance collapse of DARTS. However, no studies have explored masked autoencoders for fully unsupervised NAS.

## 3  METHOD

### 3.1  MAE-NAS: DARTS BASED ON MASKED AUTOENCODERS

Let $\mathcal{L}_{train}$ and $\mathcal{L}_{val}$ represent the training and validation loss, respectively. For DARTS Liu et al. (2019b), its goal is to find $\alpha^*$ that minimizes the validation loss $\mathcal{L}_{val}(w^*, \alpha^*)$, where the weights $w^*$ associated with the architecture are obtained by minimizing the training loss $w^* = \arg\min_w \mathcal{L}_{train}(w, \alpha^*)$. This formulation leads to a two-level optimization problem:

$$
\begin{aligned}
\min_{\alpha} \quad & \mathcal{L}_{val}(w^*(\alpha), \alpha) \\
s.t. \quad & w^*(\alpha) = \arg\min_w \mathcal{L}_{train}(w, \alpha)
\end{aligned}
\tag{1}
$$

Our approach, MAE-NAS, is grounded in a crucial observation: supervised neural architecture search often yields final models that overfit the training data. In other words, regardless of how we optimize $\alpha$ and $w$ in Equation 1, these models consistently achieve near-zero training errors. However, the ultimate goal of the search process is to identify architectures that exhibit strong generalization performance on the validation set. This presents an inherent contradiction in supervised learning. With this perspective in mind, we propose leveraging the widely-used SimMIM Xie et al. (2022) as a proxy task for NAS. In this way, we seek to discover models with enhanced generalization capabilities in the unsupervised paradigm. Building upon DARTS, the new optimization objective is formulated as follows:

$$
\begin{aligned}
\min_{\alpha} \quad & \mathcal{L}_{val}^M(w^*(\alpha), \alpha, M) \\
s.t. \quad & w^*(\alpha) = \arg\min_w \mathcal{L}_{train}^M(w, \alpha^*, M)
\end{aligned}
\tag{2}
$$

where $M$ denotes the set of masked pixels, and $\mathcal{L}_{val}^M$ and $\mathcal{L}_{train}^M$ represent the image reconstruction loss, which is identical to SimMIM (more details in appendix A.1.1). And the SSL-style objective is the sole loss function and doesn't bring in extra training cost. As shown in Figure 1, our method comprises an encoder that transforms the input into a latent representation, as well as a decoder that reconstructs the original input. Specifically, the same supernet as DARTS serves as the backbone of the encoder. In this way, the masked autoencoder becomes robust NAS learners, seeking to learn promising encoder architectures from the DARTS search space, resulting in the minimal image reconstruction loss. Note applying MAE on convolution networks is non-trivial, we generally follow SparK Tian et al. (2023) for this purpose and give the details in Appendix A.1.1.

### 3.1.1 ESCAPING FROM PERFORMANCE COLLAPSE.

DARTS exhibits a significant decline in performance when skip connections become dominant in a supervised setting. Numerous studies Chu et al. (2021a); Xu et al. (2020) have shed light on the underlying cause of this behavior. It is attributed to unfair competition between skip connections and other operations, resulting in unstable training of the supernet. Consequently, several approaches Chu et al. (2021a); Zela et al. (2020) have been proposed to address this issue by introducing various types of regularization to facilitate DARTS in escaping local optima and achieving better generalization properties. For example, R-DARTS Zela et al. (2020) seeks to add $L_2$ or *ScheduledDropPath* regularization to the optimization objective. We cannot help but ask a question: does the same issue also exist within the unsupervised paradigm?

In response to the above question, we conduct multiple independent repeated experiments based on MAE-NAS. In an unsupervised setting, we observe an interesting phenomenon where the occurrence of performance collapse is highly correlated with the size of the mask ratio. Specifically, when the mask ratio is less than 0.5, the probability of collapse is significantly high, whereas when the mask ratio exceeds 0.5, the collapse phenomenon no longer exists.

Next, we attempt to explain the phenomena. Image reconstruction in MAE is a more difficult proxy task than the supervised classification originally adopted in DARTS. And the larger the mask ratio, the more challenging it is to reconstruct the image. On the other hand, the encoder is designed to identify high-performance architectures, which have greater capability to better restore the masked image. When a large mask ratio makes image reconstruction difficult, the encoder naturally learns a more powerful architecture in order to better fulfill the optimization objective. This effectively prevents the encoder from converging to some poor architectures, which helps escape from performance collapse. Additionally, the unsupervised proxy can be also viewed as a powerful regularization where the mask ratio controls the strength of the regularization. Remarkably, the finding aligns with the conclusion of collapse in a supervised setting.

### 3.1.2 HIERARCHICAL DECODER.

At a glimpse, adjusting the mask ratio seems to be a solution. However, the encoder may inadvertently discard promising architectures if the thresholds are set inappropriately. As previously mentioned, the collapse issue is largely due to the DARTS method. The unfair competition between skip connections and other operations leads to highly unstable training in the searching stage Chu et al. (2020). To stabilize the training of DARTS and fundamentally address the problem of performance collapse, we present a more elegant solution called hierarchical decoding.

The original decoder in SimMIM takes the tokens derived from the encoder as inputs and subsequently processes them by a series of transformer blocks to reconstruct the image. In contrast, our encoder (i.e. DARTS) is designed to extract hierarchical features of different resolutions, denoted as $F_1$, $F_2$, and $F_3$, which encode fine-grained and coarse-grained information of the image. To fully supervise such features at different levels, we reconstruct the image separately from $F_1$, $F_2$, and $F_3$. Subsequently, such multi-grained outputs are aligned to the same scale by upsampling and combined with summation and linear operations, ultimately producing the reconstructed image. This process can be mathematically represented as follows:

$$I_{rec} = Linear(Conv(F_1) + \\ Upsample(Conv(F_2), 2) + Upsample(Conv(F_3), 4)) \tag{3}$$

where $I_{rec}$ represents the reconstruction image. In detail, we only focus on the part of masked patches when computing the loss, disregarding other regions of the input image. Compared to supervised NAS methods like DARTS, our approach only introduces the hierarchical decoder for image reconstruction. Such a decoder is extremely lightweight, as light as a convolution layer, which brings negligible computational cost.

The motivation behind this design encompasses two aspects: firstly, it can accelerate gradient back-propagation greatly and improve training stability effectively. The experimental results (Appendix A.2.2) show that the hierarchical decoder leads to the smoother convergence curve and the lower training loss. Secondly, it allows to learn more robust hierarchical features, enabling us to discover stronger vision backbones. Specifically, we think that single-scale algorithm cannot learn multi-scale features well. The multi-scale structure has been a universal paradigm in computer vision.

Table 1: CIFAR-10 results in DARTS search space. The average results of 5 independent experiments are reported. ‡: Avg, †: Best

| Models | Params(M) | FLOPs(M) | Top-1 Acc.(%) | Supervised | Cost(GPU days) |
|---|---|---|---|---|---|
| NASNet-A Zoph et al. (2018a) | 3.3 | 608 | 97.35 | Yes | 2000 |
| ENAS Pham et al. (2018) | 4.6 | 626 | 97.11 | Yes | 0.5 |
| DARTS Liu et al. (2019b) | 3.3 | 528 | 97.0±0.14 | Yes | 0.4 |
| GDAS Dong & Yang (2019b) | 3.4 | 519 | 97.07 | Yes | 0.2 |
| P-DARTS Chen et al. (2019a) | 3.4 | 532 | 97.5 | Yes | 0.3 |
| PC-DARTS Xu et al. (2020) | 3.6 | 558 | 97.43 | Yes | 0.1 |
| ROME-v1 Wang et al. (2023) | 4.5 | 683 | 97.5 | Yes | 0.3 |
| DARTS- † Chu et al. (2021a) | 3.5 | 568 | 97.5 | Yes | 0.4 |
| **Ours** † | 3.8 | 605 | 97.5 | No | 0.4 |
| P-DARTS Chen et al. (2019a) | 3.3± 0.21 | 540±34 | 97.19±0.14 | Yes | 0.3 |
| R-DARTS Zela et al. (2020) | - | - | 97.05±0.21 | Yes | 1.6 |
| DARTS- ‡ Chu et al. (2021a) | 3.5±0.13 | 583±22 | 97.41±0.08 | Yes | 0.4 |
| ROME-v1 Wang et al. (2023) | 4.0±0.6 | 670±21 | 97.37±0.09 | Yes | 0.3 |
| **Ours**‡ | 4.1±0.2 | 639±34 | **97.43±0.05** | No | 0.4 |

For instance, the pyramid networksLin et al. (2017a) cope with variations in object scales by its hierarchical design. Masked modeling is originally applied to transformers in a single-scale manner. Simply transferring it to convnets will lose the advantage of model hierarchy. Given that this work explores convnet-style search spaces, hierarchical decoder becomes a natural choice.

### 3.1.3 RELATIONSHIP TO PRIOR WORKS.

Label-free NAS is not new in the NAS field. Previous literature such as UnNAS Liu et al. (2020) and RLNAS Zhang et al. (2021) have demonstrated that label-free NAS can make NAS work as well as supervised NAS. To our best knowledge, we are the first to explore the MAE paradigm in the unsupervised setting, and it's not straightforward to make it work. Directly applying it suffers from the performance collapse issue like DARTS. We couple the masked autoencoder's objective with our proposed Hierarchical Decoder to address the collapse issue in DARTS and its variants. We conduct comprehensive experiments across various datasets and tasks to verify the robustness and generalization, which demonstrates MAE serves as an almost perfect proxy task for NAS. In contrast, other unsupervised proxy metrics each have their limitations and constraints in application scenarios. For example, the angle metric adopted by RLNAS does not apply to the architectures with non-parametric operators (different activation functions, max pooling, and average pooling). UnNAS seeks to introduce several unsupervised proxies (rotation, color, and jigsaw tasks) for NAS, but the experiments Liu et al. (2020) have shown that the performance of these proxy tasks is not consistent across different datasets. This has impacted its use cases and application scopes.

## 4 EXPERIMENTS

### 4.1 SEARCH SPACES AND TRAINING DETAILS

Comprehensive experiments are conducted on several popular search spaces, including NASBench-201 Dong & Yang (2020), NASBench-101Ying et al. (2019), TransNas-Bench-101Duan et al. (2021) and DARTS-based search spaces. Following the experiment settings of DARTS- Chu et al. (2021a), we apply the searching, training, and evaluation procedure on the standard DARTS search space (named $S0$). For other DARTS-like search spaces ($S1$-$S4$) proposed in R-DARTS Zela et al. (2020), we follow the same settings as the original paper. As the comparison methods, S-DARTS Chen & Hsieh (2020) differs from R-DARTS in layers and initial channels for training from scratch on CIFAR-100 Krizhevsky et al. (2009). For a fair comparison, we align such two training settings respectively. Besides, lots of ablation studies and analytical experiments are performed on NASBench-201, NASBench-101 and TransNas-Bench-101, which are built for benchmarking NAS algorithms. For ImageNet Deng et al. (2009), our method applies PC-DARTS Xu et al. (2020) to search on

Table 2: Search results on ImageNet. Models in the top block are searched on CIFAR-10 and trained from scratch on ImageNet. The rest models (also ours) are searched and trained both on ImageNet.

| Models | FLOPs(M) | Params(M) | Top-1 Acc. | Top-5 Acc. | Cost(GPU Days) |
|---|---|---|---|---|---|
| NASNet-A Zoph et al. (2018a) | 564 | 5.3 | 74.0% | 91.6% | 2000 |
| DARTS Liu et al. (2019b) | 574 | 4.7 | 73.3% | 91.3% | 0.4 |
| SNAS Xie et al. (2019) | 522 | 4.3 | 72.7% | 90.8% | 1.5 |
| PC-DARTS Xu et al. (2020) | 586 | 5.3 | 74.9% | 92.2% | 0.1 |
| FairDARTS-B Chu et al. (2020) | 541 | 4.8 | 75.1% | 92.5% | 0.4 |
| AmoebaNet-A Real et al. (2019) | 555 | 5.1 | 74.5% | 92.0% | 3150 |
| MnasNet-92 Tan et al. (2019) | 388 | 3.9 | 74.79% | 92.1% | 3791 |
| FBNet-C Wu et al. (2019) | 375 | 5.5 | 74.9% | 92.3% | 9 |
| FairNAS-A Chu et al. (2021b) | 388 | 4.6 | 75.3% | 92.4% | 12 |
| PC-DARTS Xu et al. (2020) | 597 | 5.3 | 75.8% | 92.7% | 3.8 |
| **MAE-NAS** (Ours) | 533 | 4.7 | **76.1%** | **92.8%** | 4.5 |

the standard DARTS search space, and the retraining setting follows MobileNetV3 Howard et al. (2019). For mask image modeling, the mask ratio is simply set to 0.5. The patch size of the mask is 8 and 4 for ImageNet and CIFAR datasets.

### 4.2 SEARCHING ON CIFAR-10

As shown in Table 1, regardless of whether it is the optimal or average result, the architectures found by our method perform well on CIFAR-10 Krizhevsky et al. (2009). It is worth emphasizing that our method doesn't require labels while achieving comparable even better performance compared with other supervised methods. Besides, the search cost is 0.4 GPU day, which is not higher than other methods. Such improvement is probably because the architectures found by our method have more FLOPs. But it's reasonable that models with higher flops are more likely to have better capability if flops are not constrained in free search mode.

### 4.3 SEARCHING ON IMAGENET

To thoroughly verify the effectiveness of MAE-NAS, we perform searching directly on a large-scale dataset ImageNet in search space $S0$, and train the searched model from scratch on ImageNet to evaluate its performance.

**Comparison with supervised NAS methods**. The results are shown in Table 2. Our approach, being an unsupervised approach, achieves 76.1% top-1 accuracy, which outperforms the searched models on CIFAR-10 by a clear margin. With fewer FLOPs and parameters, MAE-NAS achieves 1% higher accuracy than FairDARTS-B. Besides, MAE-NAS also stands out among all searched models on ImageNet. Compared with the supervised NAS approaches, MAE-NAS obtains the highest 76.1% top-1 accuracy with on-par FLOPs, parameters, and search cost. Such results fully demonstrate the potential of masked autoencoders as a proxy task in the NAS field.

**Comparison with unsupervised NAS methods**. To make apple-to-apple comparisons, we strictly follow the unsupervised NAS paradigm as UnNAS Liu et al. (2020) and RLNAS Zhang et al. (2021). In this paradigm, the searched models from the unsupervised searching stage are finally trained on labeled datasets to compare the performance. As shown in Table 3, compared with UnNAS and RLNAS, MAE-NAS achieves comparable even better performance with fewer FLOPs, parameters, or search costs. Here, we would like to emphasize that the relative improvements brought by MAE-NAS are comparable to some state-of-the-art (SOTA) NAS methods Chu et al. (2021a); Wang et al. (2023). It is worth mentioning that the search performance largely depends on the search space. The performance within the DARTS search space has nearly been saturated due to years of community effort. Under these circumstances, significant improvements with new approaches are very hard.

Table 3: Comparison with unsupervised NAS methods on ImageNet.[†]: rotation task, [‡]: color task. We evaluate the search cost of RLNAS by running their open-source code under the hardware environment aligned with ours.

| Method | FLOPs(M) | Params(M) | Top-1 Acc.(%) | Cost(GPU Days) |
|---|---|---|---|---|
| UnNAS [†] | 552 | 5.1 | 75.8 | - |
| UnNAS [‡] | 587 | 5.3 | 75.5 | - |
| RLNAS | 561 | 5.2 | 75.9 | 8.33 |
| **Ours** | **533** | **4.7** | 76.1 | 4.5 |

## 4.4 SEARCHING ON NAS-BENCH-201

NAS-Bench-201 Dong & Yang (2020) shares a similar skeleton as DARTS and differs from DARTS in the number of layers and nodes. Importantly, the search space trains 15625 models from scratch and provides their ground-truth performance, which allows researchers to focus on the algorithms itself without unnecessary repetitive training of searched models. As shown in Table 4, search results on NASBench-201 further verify the superiority of MAE-NAS over supervised NAS methods. First, MAE-NAS helps the native DARTS resolve the problem of collapse. Second, compared to gradient-based methods(DARTS, GDAS Dong & Yang (2019b), SETN Dong & Yang (2019a)) and non-gradient-based methods including the evolutionary search algorithm (REA Real et al. (2019)) and the random search algorithm (RSPS Li & Talwalkar (2020)), our approach sets the new state of the art on all comparison datasets, approaching the optimal solution of the search space. Besides NAS-Bench-201, more comparison experiments are conducted on TransNas-Bench-101 search space in Appendix A.2.4.

Table 4: Search results on NAS-Bench-201. The average results of 4 runs of search are reported.

| Method | Cost (hours) | CIFAR-10 | | CIFAR-100 | | ImageNet16-120 | |
|---|---|---|---|---|---|---|---|
| | | valid | test | valid | test | valid | test |
| DARTS$^{1st}$ | 3.2 | 39.77±0.00 | 54.30±0.00 | 15.03±0.00 | 15.61±0.00 | 16.43±0.00 | 16.32±0.00 |
| DARTS$^{2st}$ | 10.2 | 39.77±0.00 | 54.30±0.00 | 15.03±0.00 | 15.61±0.00 | 16.43±0.00 | 16.32±0.00 |
| GDAS | 8.7 | 89.89±0.08 | 93.61±0.09 | 71.34±0.04 | 70.70±0.30 | 41.59±1.33 | 41.71±0.98 |
| SETN | 9.5 | 84.04±0.28 | 87.64±0.00 | 58.86±0.06 | 59.05±0.24 | 33.06±0.02 | 32.52±0.21 |
| REA | - | 90.02±0.07 | 93.66±0.08 | 71.39±0.08 | 70.98±0.41 | 42.95±1.42 | 42.17±0.83 |
| RSPS | - | 83.98±0.29 | 86.46±0.02 | 57.67±0.05 | 58.93±0.26 | 32.92±0.05 | 31.25±0.19 |
| **Ours** | 3.2 | **90.63±0.31** | **93.74±0.11** | **71.42±0.07** | **71.75±0.39** | **43.17±1.11** | **43.75±0.96** |
| optimal | n/a | 91.61 | 94.37 | 73.49 | 73.51 | 46.77 | 47.31 |

## 4.5 ROBUSTNESS ON MULTIPLE SEARCH SPACES AND DATASETS

To validate the robustness of the proposed method, comparative experiments are conducted across four search spaces (S1-S4), two datasets (CIFAR-10, CIFAR-100), and multiple SOTA NAS methods. As the search process of many NAS methods is not always stable, to ensure the fairness of our experiments, we independently repeat each experiment three times and take the average results. As shown in Table 5, without labels, MAE-NAS consistently achieves comparable even better performance than supervised NAS methods on different search spaces and datasets. Taking S3 as an example, our approach discovers the model with the error rate of 16.51% on CIFAR-100, which outperforms other NAS methods with a clear margin.

## 4.6 GENERALIZATION ABILITY

The generalization ability of the proposed method is verified on downstream tasks. Specifically, we transfer different NAS models searched and pre-trained on ImageNet to the detection task for fine-tuning and evaluation. RetinaNet Lin et al. (2017b) and MS COCO dataset Lin et al. (2014) are chosen as the backbone and validation dataset. To make a fair comparison, we follow RLNAS Zhang et al. (2021) for both pre-training and fine-tuning. The only difference lies in that the backbone of

Table 5: Comparison on CIFAR-10/100 and various search spaces. The average error rate of 3 found architectures is reported. [†]: under Zela et al. (2020) settings where CIFAR-100 models have 8 layers and 16 initial channels (The best is in boldface). [‡]: under Chen & Hsieh (2020) training settings where all models have 20 layers and 36 initial channels (best in boldface).

| Benchmark | | DARTS[†] | R-DARTS[†] | | DARTS[†] | | Ours[†] | PC-DARTS[‡] | SDARTS[‡] | | Ours[‡] |
| | | | DP | L2 | ES | ADA | | | RS | ADV | |
|---|---|---|---|---|---|---|---|---|---|---|---|
| C10 | S1 | 3.84 | 3.11 | **2.78** | 3.01 | 3.10 | 2.91 | 3.11 | **2.78** | 2.73 | 2.91 |
| | S2 | 4.85 | 3.48 | 3.31 | 3.26 | 3.35 | **2.67** | 3.02 | 2.75 | 2.65 | **2.67** |
| | S3 | 3.34 | 2.93 | 2.51 | 2.74 | 2.59 | **2.49** | 2.51 | 2.53 | 2.49 | **2.49** |
| | S4 | 7.20 | 3.58 | 3.56 | 3.71 | 4.84 | **2.73** | 3.02 | 2.93 | 2.87 | **2.73** |
| C100 | S1 | 29.46 | 25.93 | 24.25 | 28.37 | 24.03 | **23.80** | 18.87 | 17.02 | **16.88** | 17.73 |
| | S2 | 26.05 | 22.30 | **22.24** | 23.25 | 23.52 | 22.55 | 18.23 | 17.56 | 17.24 | **17.12** |
| | S3 | 28.90 | 22.36 | 23.99 | 23.73 | 23.37 | 21.37 | 18.05 | 17.73 | 17.12 | **16.51** |
| | S4 | 22.85 | 22.18 | 21.94 | **21.26** | 23.20 | 21.87 | 17.16 | 17.17 | **15.46** | 16.56 |

Table 6: Object detection performance on MS COCO for the models searched in the DARTS search space. [*]: rotation task

| Method | Params(M) | FLOPs(M) | $AP(\%)$ | $AP_{50}(\%)$ | $AP_{75}(\%)$ |
|---|---|---|---|---|---|
| Random | 4.7 | 519 | 31.7 | 50.4 | 33.4 |
| DARTS | 4.7 | 531 | 31.5 | 50.3 | 33.1 |
| P-DARTS | 4.9 | 544 | 32.9 | 51.8 | 34.8 |
| PC-DARTS | 5.3 | 582 | 32.9 | 51.8 | 34.8 |
| UnNAS[*] | 5.1 | 552 | 32.8 | 51.5 | 34.7 |
| RLNAS | 5.2 | 561 | 32.9 | 51.6 | 34.8 |
| Ours | 4.7 | 533 | **33.0** | **51.8** | **35.1** |

RetinaNet is replaced with the model searched by our approach. Table 6 demonstrates that the searched model of MAE-NAS in the DARTS search space achieves higher AP on the COCO dataset than other methods.

## 4.7 SENSITIVITY ANALYSIS OF MASK RATIO AND PATCH SIZE

In MAE, mask ratio and patch size are two important parameters, which greatly affect the modeling performance. Mask ratio refers to the proportion of pixels in an image that are randomly masked or hidden during the training process. The masking operation helps the model learn robust representations by forcing it to reconstruct the original image from an incomplete or corrupted input. Patch size refers to the size of the masked patches in an image. These patches are randomly selected and masked during training, and the model is trained to reconstruct the original image based on the remaining unmasked pixels. Thus the patch size determines the spatial extent of the masked regions in an image.

Table 7: Search performance on CIFAR-10 in S0 w.r.t the mask ratio and patch size.

| Mask Ratio | Top-1 Error (%) | Patch Size | Top-1 Error (%) |
|---|---|---|---|
| 0.1 | 2.79±0.18 | 2 | 2.75±0.24 |
| 0.3 | 2.77±0.14 | 4 | **2.63±0.11** |
| 0.5 | **2.65±0.08** | 8 | 2.71±0.12 |
| 0.7 | 2.80±0.19 | 16 | 2.74±0.21 |

As shown in Table 7, we evaluate the sensitivity of MAE-NAS to mask ratio and patch size. To observe their impact more meticulously, the mask ratio is sampled at intervals of 0.2 from 0.1 to 0.7, and the patch size is sampled at intervals of 2 from 2 to 16. We find that such two parameters have a minimal impact on the final search results, which affirms the robustness of our method. Note

Table 8: Searching performance on CIFAR-10 with different mask ratios. HD is an abbreviation for Hierarchical Decoder. Each experiment is repreated three times.

|  | Mask Ratio | Top-1 Error (%) | No. of skips |
|---|---|---|---|
| Ours | 0.2 | 2.67±0.14 | 2 |
|  | 0.4 | 2.65±0.27 | 1 |
|  | 0.6 | 2.74±0.13 | 1 |
|  | 0.8 | 2.76±0.28 | 1 |
| w/o HD | 0.2 | 3.74±0.39 | 6 |
|  | 0.4 | 2.99±0.33 | 5 |
|  | 0.6 | 2.85 ±0.29 | 2 |
|  | 0.8 | 2.77±0.24 | 1 |

that MAE doesn't provide an automatic way to calibrate these hyper-parameters either. However, we empirically find that utilizing the default hyper-parameters of the MAE already suffices to search good models for our method.

### 4.8 ABLATION OF HIERARCHICAL DECODER

To evaluate the impact of the hierarchical decoder (HD), we seek to replace the HD module in our method with a regular decoder used in SimMIM Xie et al. (2022). In Table 8, we report the number of skip connections and the top-1 accuracy of found architectures in two settings. It is not difficult to observe that MAE-NAS without the HD module is greatly affected by performance collapse when the mask ratio is relatively small (i.e., smaller than 0.5), whereas our method exhibits stable performance across different mask ratios.

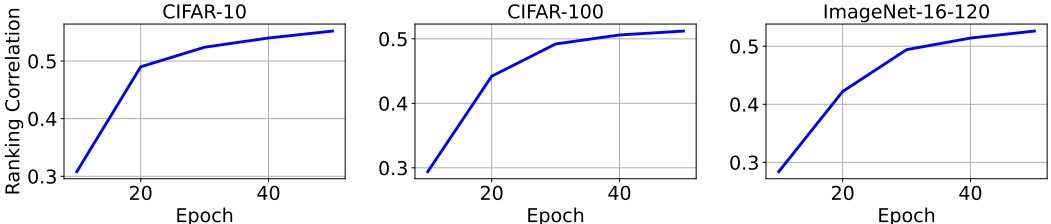

Figure 2: Ranking correlation of accuracy and image reconstruction quality on NAS-Bench-201.

## 5 IN-DEPTH ANALYSIS AND DISCUSSIONS

Strict theoretical analysis for why MAE is a good proxy is hard. Except for the extensive experimental results on standard benchmarks, we also explore the underlying mechanism in this section, which indicates the accuracy of searched architecture has a strong relation with the quality of image restoration. The better the image is restored, the better accuracy the searched architecture obtains. From this perspective, our method must learn a high-performance architecture to restore the masked image better.

### 5.1 CORRELATION BETWEEN ACCURACY AND RECONSTRUCTION QUALITY

To figure out how masked autoencoders help discover promising architectures, we begin by training a weight-sharing supernet based on NAS-Bench-201, as described in DARTS. Next, we evaluate various metrics, including ground-truth accuracy and image reconstruction score, for child models by inheriting the optimized weights from the supernet. Here, the image reconstruction score of a model is computed as the average reconstruction loss across the validation dataset multiplying $-1$. Then we rank the child models based on their image reconstruction score and ground truth accuracy respectively. Following previous works Chu et al. (2021b); Li et al. (2020b); Yu et al. (2020), the

Kendall's Tau correlation between such two sets of rankings is finally calculated to measure the correlation between ground truth accuracy and image reconstruction quality.

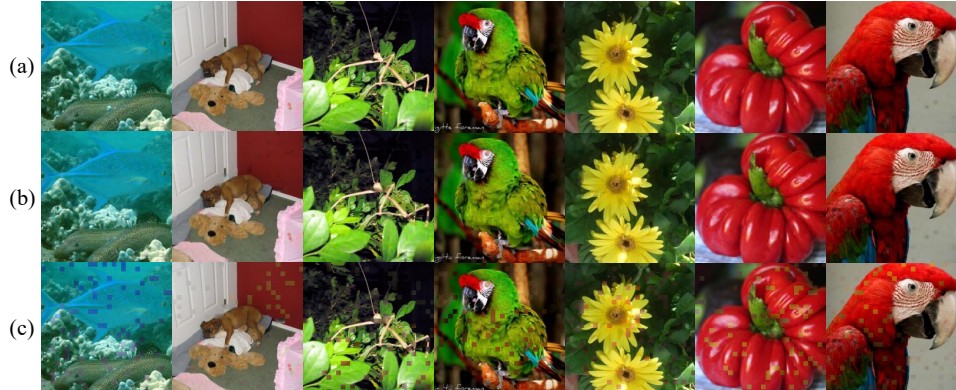

Figure 3: Comparison of the original images (a) and the reconstructed images on ImageNet. The second and third rows represent the reconstructed images of the MAE-NAS supernet under the settings of w/ HD (b) and w/o HD (c) respectively.

Figure 2 shows the ranking correlation based on supervised accuracy and reconstruction quality on three datasets (CIFAR-10, CIFAR-100, ImageNet-16-120). Prior work has demonstrated the correlation between random scores and accuracy is less than 0.01 (Table 2 of ABS Hu et al. (2020)). So the reported correlation sufficiently indicates a strong association between the reconstruction score and accuracy. To further substantiate this conclusion, we follow UnNAS and report Spearman's rank correlation between the proxy task and the image classification task in the form of the scatter plot on NASBench-101Ying et al. (2019) in appendix A.2.3. We believe image reconstruction based on masked autoencoders is indeed a promising proxy task for NAS and effectively explains why MAE-NAS works well across different search spaces and datasets.

### 5.2 VISUALIZATION OF IMAGE RECONSTRUCTION

In Section 3.1, we explain from the view of optimization why HD is able to solve the issue of performance collapse: it accelerates gradient back-propagation greatly and improves training stability effectively. We also draw the training loss during the search phase in Appendix A.2.2, and find that HD leads to the smoother convergence curve and the lower training loss than the one without it.

We rethink this question from the view of image reconstruction. As shown in Figure 3, MAE-NAS achieves superior image reconstruction quality over its counterpart without HD. Conceptually, the encoder is expected to be equipped with high-performance architectures, which have stronger ability to better restore the masked image. If the reconstruction image has higher quality, in order to achieve the goal, the encoder naturally learns a more powerful architecture. Such mechanism prevents the encoder from converging to a poor architecture, thus helps MAE-NAS escape from performance collapse.

## 6 CONCLUSION

Obtaining labeled data is time-consuming, making unsupervised NAS methods attractive. We propose MAE-NAS based on Masked Autoencoders that eliminates the need for labeled data. Our approach replaces the supervised learning objective with a reconstruction loss, enabling the discovery of network architectures with stronger representation and improved generalization. Experimental results on seven search spaces and three datasets demonstrate the effectiveness of MAE-NAS, achieving comparable performance with its counterparts under the same complexity constraint. Such experiments primarily covers image understanding tasks such as classification and object detection. However, it's nontrivial to apply our method to image generation tasks, and we'll explore this in the future work.

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

# A  APPENDIX

## A.1  IMPLEMENTATION DETAILS

### A.1.1  APPLYING MAE ON CONVOLUTIONAL NETWORKS

We follow SimMIM Xie et al. (2022) and SparK Tian et al. (2023) to apply MAE on convolutional networks. In details, Masked Autoencoders (MAE) captures representations by employing a masked image modeling technique that conceals parts of the input image signals and reconstructs the true signals in the obscured zones. This system is built upon three key components:

- Masking strategy: the patch-aligned random masking strategy is adopted. We process images in patches, making it practical to apply masking at the patch level, where a patch is entirely exposed or completely concealed. MAE-NAS adopt $8 \times 8$ and $4 \times 4$ as the default masked patch size of ImageNet and CIFAR datasets respectively.

Table 9: P-DARTS and its combination with ours on CIFAR-10. The manual tricks are removed in our experiments.

| Method | Setting | Top-1 Acc. (%) |
|--------|---------|----------------|
| P-DARTS | w/o tricks | 96.48±0.55 |
| MAE-NAS | w/o tricks | 97.16±0.14 |

Table 10: The combination of PC-DARTS and ours on CIFAR-10. Searching is repeated three times.

| Method | Top-1 Acc. (%) | Cost |
|--------|----------------|------|
| PC-DARTS | 97.09±0.14 | 3.75h |
| MAE-NAS | 97.27±0.18 | 3.41h |

- Encoder architecture: this component is responsible for deriving a latent feature representation from the masked image, which serves as the basis for predicting the original signals in the masked region. MAE-NAS adopts the DARTS search space as the backbone.

- Decoder and prediction target: the decoder adopt the HD module and it is utilized on the latent feature representation to generate the original signals within the masked region. Prediction target defines the form of original signals to predict. And the $l_1$-loss is employed on the masked pixels:

$$L = \frac{1}{\Omega(x_M)}||y_M - x_M||_1$$

where $x, y \in \mathbb{R}^{3HW \times 1}$ are input RGB values and the prediction values respectively. The symbol $M$ represents the collection of pixels that have been masked, while $\Omega(\cdot)$ indicates the count of elements.

## A.2 More Experimental Results

### A.2.1 Combination with Other Variants

We verify the power of our approach combined with other NAS methods. We choose two popular NAS algorithms (P-DARTS and PC-DARTS) as baselines, whose codes are available, to apply MAE as their NAS proxy for further improvements. All experiments are conducted on ImageNet, whose training set is split into two parts: 50,000 images for validation and the rest for training.

**P-DARTS** The motivation behind P-DARTS Chen et al. (2019a) is to close the depth gap between searching and training by presenting a progressive search strategy. The method starts with a small network and progressively increases its size and complexity over multiple stages. Meanwhile, some prior knowledge is introduced for search space regularization, to avoid the issue of performance collapse. For example, they apply dropout after each skip-connect operation. Besides, they control the number of preserved skip-connects manually. The aforementioned strategies, to some extent, compromise the fairness of the comparison. To this end, we remove these artificial limitations for fair comparison. We run P-DARTS without handcrafted tricks and our approach each three times to get an average result. As shown in Table 9, our approach achieves 97.16% Top-1 accuracy, which is 0.68% higher than P-DARTS. In conclusion, our method effectively addresses the problem of performance collapse for P-DARTS without human prior.

**PC-DARTS** The motivation behind PC-DARTS Xu et al. (2020) is to deal with the challenge of memory and computational efficiency of NAS. Traditional methods for architecture search require a large number of parameters and operations, making them computationally expensive and memory-intensive. To this end, PC-DARTS proposes using partial channel connections and allows for parameter sharing across different channels in a convolutional neural network, which effectively reduces the number of parameters and the computational cost.

To verify the effectiveness of masked autoencoders as a NAS proxy in the PC-DARTS setting, we compare the performance of PC-DARTS with its combination with ours. To ensure the reproduction of our results, we utilize their source code and conduct multiple repeated experiments with different random seeds under the same settings. As shown in Table 10, MAE-NAS achieves a 0.18% top-1 accuracy increase with the lower search cost compared to PC-DARTS.

Overall, the above results demonstrate the potential of MAE-NAS to enhance the performance of existing NAS algorithms, even under suboptimal configurations. We believe that our approach can

be further optimized and applied to a wider range of scenarios in the NAS field, paving the way for more efficient and effective neural architecture search paradigms.

### A.2.2 CONVERGENCE CURVE OF MAE-NAS

We attempt to compare the convergence curve of training loss on ImageNet for MAE-NAS in two

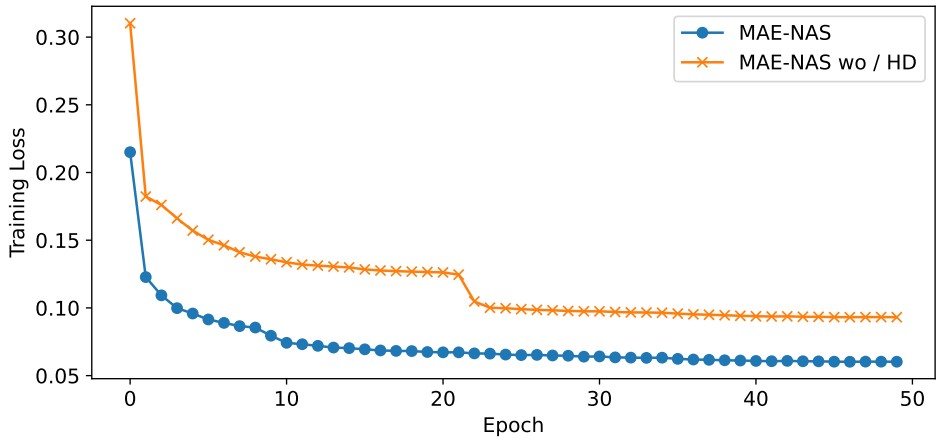

Figure 4: The convergence curve of training loss on ImageNet.

settings: with and without HD. The results in Figure 7 show that the inclusion of the HD module leads to the lower loss for MAE-NAS and the smoother convergence curve.

### A.2.3 CORRELATION BETWEEN PRETEXT TASK AND HE TASK OF IMAGE CLASSIFICATION

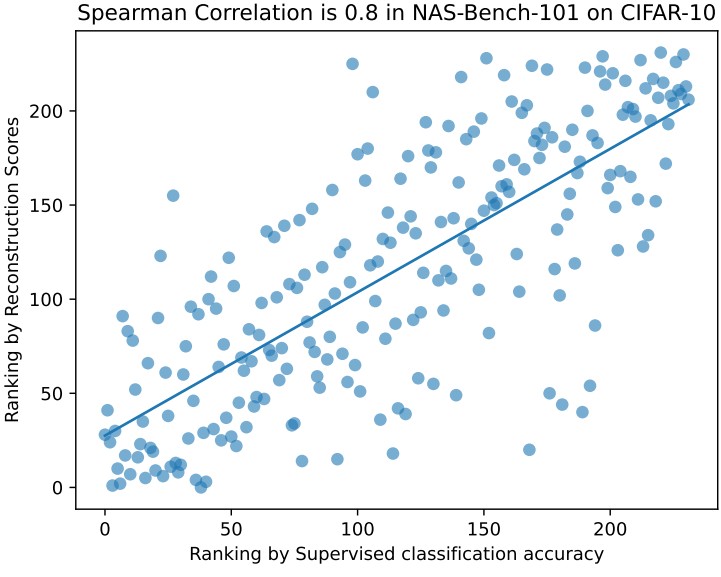

Figure 5: The ranking correlation between the scores of the pretext task and the accuracy of image classification in NAS-Bench-101 on CIFAR-10.

### A.2.4 SEARCHING COMPARISON EXPERIMENTS ON TRANSNAS-BENCH-101.

As in Table 11, both RS and REA achieve comparable even better performance with MAE proxy

Table 11: Comparison with RS and REA on TransNas-Bench-101.

| Method | Proxy | Supervised | Cls Acc.(%) | Seg.(mIoU) |
|---|---|---|---|---|
| RSBergstra & Bengio (2012) | accuracy | Yes | 45.16±0.4 | 25.21±0.4 |
| RS | mae | No | 45.34±0.4 | 25.35±0.4 |
| REAReal et al. (2019) | accuracy | Yes | 45.39±0.2 | 25.52±0.3 |
| REA | mae | No | 45.47±0.3 | 25.77±0.3 |

to the original methods. This proves that, as an unsupervised metric, MAE proxy is very promising.

### A.2.5 GENOTYPE VISUALIZATION OF SEARCHED MODEL

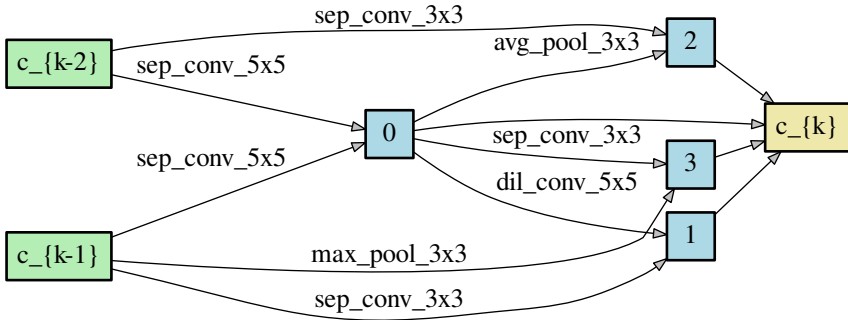

Figure 6: The normal cell searched on ImageNet.

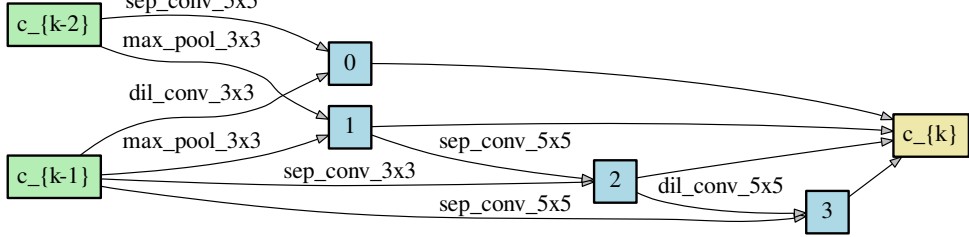

Figure 7: The reduction cell searched on ImageNet.

