# OpenReview forum: "Powering Neural Architecture Search with Robust Masked Autoencoders"
_ICLR.cc/2025/Conference — Submitted to ICLR 2025_

### Official Review · Reviewer_jWWf · 2024-11-02

**Soundness:** 3
**Presentation:** 3
**Contribution:** 2
**Rating:** 5
**Confidence:** 3

**Summary:**

This paper presents a new neural architecture search method based on unsupervised mask autoencoder. The author specifically designed their method to address performance collapse issues with adjustments of the mask ratio and a curated hierarchical decoder. They further validated their proposed approaches on cifar10, imagenet and nas bench 201 etc..

**Strengths:**

This presents a clear and straightforward way of leveraging unsupervised mae for neural architecture search, and it's interesting to understand how they synergize in the NAS framework.

**Weaknesses:**

1. While the idea of leveraging MAE for neural architecture search is straightforward, there is a lack of discussion and understanding of how they synergize together (more can be added in Sec 3). For example, it's vague whether there are any characteristics making MAE suitable for this task and a better choice over other unsupervised NAS methods. It's hard to tell the advantage from unsupervised NAS comparisons in Sec 4.

2. Regarding the hierarchical decoder, the authors have compared against the use of a regular decoder in Sec 4.8, still some doubts why this particular decoder design works the best. Also, as part of the novelty is from this hierarchical decoder (while combining mae and nas is more straightforward), I'd expect improvements from this to be more evident. However, the effectiveness of the proposed decoder is limited with high mask ratio, which is eventually what the authors adopt to use.

**Questions:**

Please see weaknesses.

---

> ### Author Response · Authors · 2024-11-25
>
> **Q1**: There is a lack of discussion and understanding of how NAS and MAE synergize together.
>
> **A1**: The motivation comes from a crucial observation: supervised neural architecture search often yields the final models that overfit the training data. In other words, regardless of how we optimize $alpha$ and $w$ in Equation 1, the searched models consistently achieve near-zero training errors.
> However, the ultimate goal of NAS is to identify architectures that exhibit strong generalization performance on the validation set. This presents an inherent contradiction in supervised NAS. With this perspective in mind, we propose leveraging MAE as the proxy task for NAS. In this way, MAE-NAS seeks to discover models with enhanced generalization capabilities in the unsupervised paradigm.
>
> our work is the first to explore the MAE paradigm for unsupervised NAS. In theory, it can serve as the proxy task for any NAS methods.
>
> **Q2**: Some doubts why this particular decoder design works the best.
>
> **A2**: Please refer A3 to Reviewer PEu7.

---

### Official Review · Reviewer_pDHk · 2024-11-03

**Soundness:** 2
**Presentation:** 2
**Contribution:** 2
**Rating:** 3
**Confidence:** 5

**Summary:**

This paper presents a novel NAS method called MAE-NAS leveraging masked autoencoders to achieve unsupervised searching. Specifically, the proposed MAE-NAS method is designed based on the optimization objectives of DARTS, along with the encoder designed based on the supernet of DARTS and the hierarchical decoder. Moreover, the proposed MAE-NAS method also has the ability to escape from performance collapse via changing the mask ratio. The proposed MAE-NAS method is evaluated on various search spaces and datasets, and the experimental results demonstrate the effectiveness of the proposed method.

**Strengths:**

1) The idea of proposed MAE-NAS method is simple and easy to understand.
2) The framework and core components of the proposed MAE-NAS method is clearly presented in the paper.
3) The performance collapse is an important problem in differentiable NAS methods, and this problem is not well explored in an unsupervised manner.

**Weaknesses:**

1) The motivation for adopting masked autoencoders to achieve unsupervised NAS is relatively unclear. In particular, the authors stated that the motivation for using masked autoencoders is that no studies have explored this direction. However, the necessity and the advantage for using masked autoencoders are still unclear. Besides, the proposed method seems a direct combination of the masked autoencoder and DARTS. I suggest the authors provide more discussions for adopting the masked autoencoders for unsupervised NAS.
2) The proposed MAE-NAS method is combined with DARTS, PDARTS, and PCDARTS to show its effectiveness in the plug-and-play manner. However, these differentiable NAS methods are relatively old. Why not integrate the proposed method into the recent differentiable NAS method to achieve the state-of-the-art performance?
3) Lack of in-depth analysis in terms of the performance collapse. Specifically, the authors states that the collapse phenomenon still exists in the unsupervised manner. However, they failed to explain why this phenomenon appears. Please provide more analysis and evidences in terms of the reason why this phenomenon can appear in the unsupervised manner.
4) Some claims are not well supported in terms of the performance collapse. For example, it is not well supported that the encoder can learns a more powerful architecture when the mask ratio becomes larger. Besides, why the unsupervised proxy can be viewed as the regularization? More empirical or theoretical evidences are needed for the points mentioned above.
5) The experimental results need further explanations. For example, the experimental results in Table 7 seem strange. When the mask ratio is set to 0.1 and 0.3, the Top-1 Errors are 2.79 and 2.77. However, when the mask ratio is set to 0.7, the Top-1 Error changes to 2.80. This phenomenon is conflict to the claim “when the mask ratio is less than 0.5, the probability of collapse is significantly high” in Section 3.1.1. Besides, it seems that when the mask ratio is small, the derived architecture does not suffer from serious performance degradation as shown in Table 8.
6) The writing needs improvements. For example, the format of the references in the main article seems incorrect. It is suggest to use “\citep{}” instead of “\cite{}”. Besides, the words “(i.e. DARTS)” should be “(i.e., DARTS)” in line 197.

**Questions:**

Please see weaknesses. Because there are a lot of ambiguous claims and weird experimental results in this version, and the motivation and novelty are not properly presented, I tend to recommend rejection temporarily.

---

> ### Author Response · Authors · 2024-11-25
>
> **Q1**: The motivation for adopting masked autoencoders to achieve unsupervised NAS is unclear.
>
> **A1**: Please refer A1 to Reviewer jWWf.
>
> **Q2**:  The methods combined with mask autoencoders are old.
>
> **A2**: DARTS is the pioneering work of differentiable NAS, and many subsequent methods are its variations. This method is clean and elegant, and combination with it can fully reflects the true effect of the proposed proxy task. Additionally, we conducted detailed comparison with recent SOTA differentiable NAS including supervised and unsupervised methods in Table 2 and Table 3, which demonstrate our approach is competitive enough in terms of performance.
>
> **Q3**: Some claims are not well supported in terms of the performance collapse.
>
> **A3**: The root cause of performance collapse lies in the design of the search algorithm and the search space, and the specific reasons have been widely studied before. It is attributed to unfair competition between skip connections and other operations, resulting in unstable training of the supernet. Additionally, the experiments in Table 8 can support our claim.
>
> **Q4**: This phenomenon is conflict to the claim “when the mask ratio is less than 0.5, the probability of collapse is significantly high”.
>
> **A4**: This is not contradictory. The experiments in Table 7 are conducted with HD enabled by default, so there is no performance collapse issue at all, and the test error is within the normal fluctuation range. In Table 8, at the setting of the small mask ratio without HD, the training error and the number of skip connections both increase, which is a clear signal of training collapse.
>
> **Q5**:  The writing needs improvements.
>
> **A5**: Thanks! We will fix the format of the references in the revised version.

---

> > ### Comment · Reviewer_pDHk · 2024-11-27
> >
> > Thanks for the feedback from authors. However, my previous concerns are not fully addressed. Specifically:
> >
> > 1) The motivation for using MAE is still not really convincing to me. The authors claimed that “supervised NAS overfits the training data”. However, I cannot find any evidence in the rebuttal or submitted paper.
> > 2) In my opinion, if the proposed method cannot be integrated into SOTA differentiable NAS methods beyond DARTS, PDARTS, and PCDARTS, the contribution of this work will be limited. Besides, the compared methods in Table 2 are still relatively old.
> > 3) As shown in Table 8, when the number of “skip connection” reaches 6, the test error is 3.74%, without significant performance drop, i.e., performance collapse. Besides, please provide more evidence for this point (at least some references I suppose).
> > 4) The point 4 in Weaknesses Section has not been addressed. Why the unsupervised proxy can be viewed as the regularization?
> >
> > I decide to keep my initial rating.

---

### Official Review · Reviewer_PEu7 · 2024-11-04

**Soundness:** 2
**Presentation:** 3
**Contribution:** 3
**Rating:** 5
**Confidence:** 4

**Summary:**

This paper addresses the challenge of Neural Architecture Search (NAS) dependency on supervised signals, which can be costly due to the need for labeled data. To overcome this, the authors propose a novel NAS approach that integrates DARTS with a masked autoencoder (MAE) paradigm, adapting it for unsupervised learning. A key challenge is the collapse of DARTS in unsupervised settings, which is mitigated by increasing the mask ratio and introducing a hierarchical decoder for enhanced performance. The method achieves comparable results to both supervised and unsupervised NAS counterparts, validated through experiments on seven datasets.

**Strengths:**

Originality and Quality: This paper explores a novel integration of MAE with DARTs, proposing that reconstruction targets can also serve as effective guidance for neural architecture search. The paper is of acceptable quality, with experiments conducted across several common benchmarks and supported by multi-run results, providing a reasonable foundation for the claims made.

Clarity: The paper is presented with straightforward figures and tables and the writing is easy to follow.

Significance: While the paper offers a fresh perspective, its significance is a primary concern, as discussed in the weaknesses section

**Weaknesses:**

Firstly, while I acknowledge that directly combining MAE and DARTS is not entirely novel, this paper falls short of fully realizing the impact of such a combination. In terms of results, as noted by the authors in Line 323, performance on large-scale datasets like IN-1K and standard benchmarks like CIFAR appears saturated. Without a clear performance or efficiency advantage, the approach lacks significance. Specifically, MAE’s strengths lie in efficiency (training speed) and scalability (e.g., training ViT-Huge on IN-1K alone), which other unsupervised methods lack. If applied effectively to NAS, I would expect MAE to offer fresh insights, such as addressing overfitting in NAS or demonstrating scalability. Unfortunately, the current work does not meet these expectations.


Secondly, several claims in the paper are ambiguous and require further clarification. For instance, the authors emphasize that directly applying MAEs with DARTS leads to training collapse, which can be mitigated by adjusting the mask ratio. However, Table 7 does not substantiate this claim, nor can I find any supporting experiments elsewhere in the paper to confirm training collapse. Additionally, while the authors propose HD to address this issue, the ablation studies indicate that HD has minimal impact on performance.


Minor: The format of the citation should be revised.

**Questions:**

1.  Why MAE+DARTS? The motivation is fundamental for this paper to improve its overall quality, given the technical novtiy is not significant and improvements are marginal. I expect to see more insight into the motivation.

2.  What is the design logic behind the proposed HD? I cannot see a clear improvement using this block.

3. What is the baseline in the paper? I cannot find a comparable method that can justify the proposed method's novelty and significance.

---

> ### Author Response · Authors · 2024-11-25
>
> **Q1**: Technical novtiy is not significant and improvements are marginal.
>
> **A1**: Please refer A1 and A2 to Reviewer qtpr.
>
> **Q2**: Supporting experiments to confirm training collapse; How HD addresses this issue.
>
> **A2**: Training collapse is a long-standing issue in the DARTS-based methods. Prior to our work, many methods such as PDARTS and RLNAS have been proposed to address the problem. Specifically, the algorithms will converge to the local optima and the searched architectures have lots of skip connection operations. As shown in Table 8, MAE-NAS without the HD module leads to serious performance collapse (more than 5 skip connections) in the setting of small mask ratio. By increasing the mask ratio or applying HD, the training collapse issue can be resolved.
>
> **Q3**: What is the design logic behind the proposed HD?
>
> **A3**: In depth, we think that single-scale algorithm cannot learn multi-scale features well. The multi-scale structure has been a universal paradigm in computer vision. For instance, the pyramid networks cope with variations in object scales by its hierarchical design. Masked modeling is originally applied to transformers in a single-scale manner. Simply transferring it to convnets will lose the advantage of model hierarchy. Given that this work explores convnet-style search spaces, hierarchical decoder becomes a natural choice. On the one hand, it helps stabilize training by enhancing gradient propagation. In Figure 3 (supp.), the hierarchical decoder leads to the smoother convergence curve and the lower training loss. On the other hand, it allows to learn more robust hierarchical features without intensive hyper-parameters tuning, enabling us to discover stronger vision backbones. In Table 8, the searched models achieve the lower Top-1 error at different settings by applying hierarchical decoder.
>
> **Q4**: Minor: The format of the citation should be revised.
>
> **A4**: Thank you. We will fix the format of the references in the revision version.

---

### Official Review · Reviewer_qtpr · 2024-11-05

**Soundness:** 2
**Presentation:** 2
**Contribution:** 2
**Rating:** 3
**Confidence:** 4

**Summary:**

This paper proposes a new DARTS based architecture search method with MAE as the objective. This enables effective NAS without supervised vision data. The results improve over existing supervised NAS methods and Unsupervised previous works.

**Strengths:**

1. The proposed method looks straight forward given the possibility of UnNAS and the effectiveness of MAE.
2. Experiments were carried out comprehensively in traditional NAS settings for convolutional neural networks.
3. Interesting phenomenon has been observed, such as the correlation between task difficulty and performance collapse, thus leading to the proposal of hierarchical decoder, etc.

**Weaknesses:**

1. Given the successful search for various pretext tasks in UnNAS, as well as the successful application of MAE as a self-supervised learning method, it is expected that MAE-NAS is possible and will generate reasonably good results. It is true that this paper shows a successful implementation of the idea, but the novelty or contribution to the field may not be sufficient for ICLR.
2. As shown in UnNAS paper, the performance difference in the architectures searched with different objectives are quite small. This conclusion also transfers to this paper where the performance gain on top of existing methods has been quite limited as shown in Table 1, 2, 3.
3. MAE was proposed for transformers and enabled efficient pre-training of transformers without labels. However, MAE-NAS works for convolutional search space only and has not shown promise in searching in more interesting spaces such as transformers. Similarly, the adoption of the convolutional space defeats the purpose of MAE paradigm of pre-training efficiency — the resulting convolutional architecture has to be trained without MAE.
4. minor: Presentation. Section 3 includes 1 subsection 3.1 only.

**Questions:**

1. Is it possible to design a new search space on top of MAE training so that masked patches can still be dropped during pre-training?
2. Does the method apply to other domains such as 3D networks or medical imaging where the best architectures have been less exploited and where data may be more limited than 2D images? This may help demonstrate the strength of the paper.

---

> ### Author Response · Authors · 2024-11-25
>
> **Q1**: The novelty or contribution may not be sufficient.
>
> **A1**: To our best knowledge, we are the first to explore the MAE paradigm for NAS in unsupervised setting, and it's not straightforward to make it work. Direcly applying it suffers from the performance collapse issue like DARTS. We couple the masked autoencoder's objective with our proposed Hierarchical Decoder to address the collapse issue in DARTS and its variants (Table~8).
>
> While other unsupervised proxy metrics each have their limitations and constraints in application scenarios, our method is neat and doesn't have these limitations.
>
> **Q2**:  The performance gain on top of existing methods has been limited.
>
> **A2**:  The relative improvements brought by MAE-NAS are comparable to the SOTA NAS methods (RLNAS and DARTS-). Moreover, we would like to emphasize that the performance of NAS algorithms depends on the search space. After a period of development, existing methods have nearly reached the upper limit of performance within the DARTS search space. Under such circumstances, we believe that significant improvements are unrealistic.
>
> In addition, various indicators such as Hessian eigenvalues have been proposed as a signal to stop searching before the performance collapses. However, these indicator-based methods tend to easily reject good architectures if the thresholds are inappropriately set. In contrast, we propose a new unsupervised paradigm, which is more  elegant. Beyond addressing the issue of performance collapse, the primary advantage of our approach is that it requires no labels during the search phase. This significantly 060 broadens the potential use cases and the scope of MAE-NAS.
>
> **Q3**: MAE-NAS works for convolutional search space only and has not shown promise in searching in transformers spaces.
>
> **A3**: We emphasize that this work doesn't care the problem of pre-training efficiency. Our motivation is to discover superior model architectures by taking MAE as the proxy task of NAS. Coincidentally, the mainstream search spaces are primarily based on CNNs. Prior to this, Spark has demonstrated that MAE can be effectively applied to CNNs and performs well.
>
> **Q4**: Presentation. Section 3 includes 1 subsection 3.1 only.
>
> **A4**: Thanks! We will fix the issue in the revision.
>
> **Q5**: Is it possible to design a new search space on top of MAE training so that masked patches can still be dropped during pre-training?
>
> **A5**: In our implementation, the positions of masked patches are simply set to 0, which is actually equivalent to discarding these patches like the transformer. To further make the convolution efficient, these masked patches can be completely discarded without participating in any computations by applying sparse convolution. We will leave it as the future work.
>
> **Q6**: Does the method apply to other domains such as 3D networks ?
>
> **A6**: Currently, we have only conducted experiments on 2D CNNs and have not yet performed experiments on 3D networks. We believe it is promising and will consider it as the future work.

---

### Meta-Review · Area_Chair_sjeP · 2024-12-20

**Metareview:**

All reviewers are on the rejecting side for the paper. The AC checks all the materials, and while appreciating the responses from the authors, the AC resonates with the reviewer consensus that the paper is below the bar for acceptance. Specifically, the significance of the work relies on the synergy between MAE and NAS, but the search space for MAE-NAS is limited to convolutional designs (not Transformers which fare well with MAE). The authors also mentioned about the work *does not care* about training efficiency, but one of the biggest advantages of MAE over supervised learning is its training efficiency as it couples well with high mask-ratio token dropping. These suggest the current exploration is not well-planned or well-executed. The authors are encouraged to rethink about the work and make further improvements.

**Additional Comments On Reviewer Discussion:**

- Reviewers are not satisfied by the motivation of using MAE for NAS merely due to overfitting with supervised learning. The AC concurs and suggests other advantages of MAE to be exploited, e.g. efficiency.

---

### Decision · Program_Chairs · 2025-01-22

Reject